# Economic burden of the persistent morbidity of nodding syndrome on caregivers in affected households in Northern Uganda

Lugala Samson Yoane Latio[1]☯, Nguyen Hai Nam[2,3]☯, Jaffer Shah[2,4], Chris Smith[1,5], Kikuko Sakai[6], Kato Stonewall Shaban[7], Richard Idro[8], Nishi Makoto[9], Nguyen Tien Huy[10]*, Shinjiro Hamano[11], Kazuhiko Moji[1]*

1 School of Tropical Medicine and Global Health, Nagasaki University, Nagasaki, Japan, 2 Online Research Club (http://www.onlineresearchclub.org), Nagasaki, Japan, 3 Division of Hepato-Biliary-Pancreatic Surgery and Transplantation, Department of Surgery, Graduate School of Medicine, Kyoto University, Kyoto, Japan, 4 Drexel University College of Medicine, Philadelphia, Pennsylvania, United States of America, 5 London School of Hygiene and Tropical Medicine, London, United Kingdom, 6 Liberal Arts and Science, Kanazawa Seiryo University, Kanazawa, Japan, 7 Department of Environment and Natural Resources, Faculty of Agriculture and Environment Gulu University, Gulu, Uganda, 8 Department of Pediatrics and Child Health, College of Health Sciences, Makerere University, Kampala, Uganda, 9 Graduate School of Asian and African Area Studies, Kyoto University, Kyoto, Japan, 10 Department of Clinical Product Development, Institute of Tropical Medicine (NEKKEN), School of Tropical Medicine and Global Health, Nagasaki University, Nagasaki, Japan, 11 Department of Parasitology, Institute of Tropical Medicine (NEKKEN), Nagasaki University, Nagasaki, Japan

☯ These authors contributed equally to this work.
* moji-k@nagasaki-u.ac.jp (KM); tienhuy@nagasaki-u.ac.jp (NTH)

**Data Availability Statement:** All relevant data are within the manuscript and its Supporting Information files.

## Abstract

### Background

Nodding syndrome (NS), is an unexplained form of epilepsy which leads to stunted growth, cognitive decline, and a characteristic nodding of the head. Current data about its impact on households in Uganda is scarce. Therefore, this study aims to assess the economic burden of the persistent morbidity of NS on caregivers in affected households in Northern Uganda.

### Methods

A cross-sectional cost-of-care study was conducted from January 2019 to February 2019 in Lakwela village-Northern Uganda in 14 households, who are members of a community-based organization (CBO) established in the village with the support of a Japanese research team, (Uganda-Japan Nodding Syndrome Network). Data was collected through question-naires. Both direct (medical and non-medical) and indirect (informal care) costs of caregiving were assessed. Indirect costs were valued using the human-capital method as loss of production.

### Results

Direct costs constituted a higher proportion of costs for NS households, accounting for on average 7.7% of household expenditure. The annual weighted mean cost per NS patient was estimated at 27.6 USD (26.4 USD direct costs, 96.2% and 1.2 USD indirect cost,

**Funding:** The author(s) received no specific funding for this work.

**Competing interests:** The authors have declared that no competing interests exist.

**Abbreviations:** ACNS, Alliance for Community with Nodding Syndrome; ADL, Activities of daily living; CBO, Community-based organization; GDP, Gross domestic product; HDL, Household activities of daily living; IADL, Instrumental activities of daily living; iVICQ, iMTA Valuation of Informal Care Questionnaire; NS, Nodding syndrome; OAE, Onchocerciasis-Associated Epilepsy; SD, Standard deviation.

3.8%). Average time spent on informal caregiving was 4.4 ±1.7 (standard deviation) hours/ week with an estimated annual informal caregiving cost of 24.85 USD and gross domestic product (GDP) loss of 412.40 USD.

## Conclusion

Direct costs due to NS are still high among households in this study. More studies are needed to investigate measures that could help bring down these costs and equally reduce the day-to-day disruption of caregiver's activities; consequently, improving the lives of these affected households and communities.

## Introduction

Nodding syndrome (NS) is a neurological condition with unknown etiology affecting previously healthy children around the ages of 3–18 [1]. It is mainly characterized by head nodding episodes with or without generalized epileptic seizures, cognitive impairment, and abnormal growth & development [2,3]. Moreover, hallucinations and other extreme forms of behaviors typical of causing self-harm have been reported [4]. The seizures are triggered when the patients consume local food [5], walk, wake up from sleep or are exposed to cold weather, although seizures may occur without any specific trigger [6,7]. NS mainly occurs in onchocerciasis-endemic areas [8,9] and some epidemiological studies have suggested its association with onchocerciasis[10–13], as well as its common traits with Nakalanga syndrome [11,14]. Thus, NS is considered a part of the spectrum of Onchocerciasis-Associated Epileptic (OAE) disorders [10].

In terms of the overall disease burden, except for Uganda, where cases were estimated at 2,083, deaths at 128 as of March 2018 [15] and a prevalence of 6.8 per 1000 children [1], the actual burden has not been established to-date in Tanzania and South Sudan [16–18]. NS is reported to impact a high socioeconomic burden for the households and communities where it occurs; for example, such patients are often reported to drop out of school and also, when left untreated, they may not be able to contribute to family income in terms of working, as well as other household duties [19]. Similarly, these patients are reported to mainly die of secondary causes like drowning, falling in fire, and at times from simply wondering off [20,21]. Therefore, parents/guardians of patients must be around to take care of them to prevent such instances. Furthermore as reported by Nakigudde et al. [22], more than one child may be affected in a household, leading to a high burden of care, which in turn overwhelms, the family's available resources. In fact, looking after children affected by NS has been equated to a full-time, unpaid occupation that is both physically and economically draining to the caregivers. As stated by Deogratius et al., because of the long-term care, there is inevitably less time for families to engage in other economic activities such as farming and thus, most households go bankrupt [23]. This situation is further exacerbated by the fact that NS occurs in poverty prone areas. For instance, for the case of Uganda, NS occurs in the northern region. Having undergone a civil war in the past, the north is the most affected region in the country suffering from a double burden of high poverty levels [24,25] and the NS epidemic [26]. It is, therefore, not surprising that low household income has been reported as one of the factors associated with NS [27].

While many studies have been performed assessing the economic burden in similar diseases like epilepsy, no such study has been done on NS (one notable economic study on NS has been

a modeling study on resource allocation on how patients could be enrolled and followed up in a nutritional program in the most resource efficient way [21]. There is, therefore, an increasing need to assess the economic burden of this novel condition on such households so as to guide policy makers on how best to support such affected communities in the most resource efficient way in order to alleviate the burden imposed on them by NS.

## Methods

### Study population

This research was a cross-sectional study that was carried out from January 16[th], 2019 to February 20[th], 2019 in Lakwela Village in Gulu district, a region in Northern Uganda. Gulu District is one of the districts that constitutes the Acholi sub-region and is estimated to be about 340 kilometers by road, north of the country's capital city, Kampala. The district's population is estimated at 275,613 [28], with about 323 (15.5% of all NS) cases officially reported between January 2012 and March 2018 [15]. The study participants were caregivers of NS patients. The inclusion criteria consisted of the households with nodding syndrome patients, who are members of a community-based organization called Alliance for Community with Nodding Syndrome (ACNS), which is being supported by Uganda-Japan Nodding Syndrome Network. After screening, we identified only 19 households who meet the inclusion criteria in this village and among them, five were excluded due to translocation to other districts. There were finally 14 households who enrolled to our study. All NS cases (according to the World Health Organization case definition of NS) in the selected households were considered as well as each household's acceptance to take part in the study.

### Data collection and tools

A structured interviewer-administered questionnaire was used and included questions on: (a) socio-demographic characteristics of both caregivers and receivers, (b) employment and work-related situations of the caregivers, (c) direct medical costs [*registration*, *consultations*, *diagnostic tests*, *medications and hospitalization char*ges], and (d) direct non-medical costs [*transportation*, *accommodation and feeding costs*]. Questions in these sections were mainly adapted from the demographic and health survey questionnaires [25]. Activities of daily living (ADL), instrumental activities of daily living (IADL), as well as household activities of daily living (HDL) of caregivers were assessed using the 'monetary valuation of informal care using the opportunity cost and proxy good methods sections' of the iMTA Valuation of Informal Care Questionnaire (iVICQ) [29]. However, the HDL and ADL questions have been adapted to the Ugandan situation based on the Oxfam survey in the Philippines, Uganda, and Zimbabwe [30]. A detailed description of the ADL, IADL and HDL activities is presented in Table 1.

Seizure severity, as perceived by the parent or caregiver, was assessed using the Hague seizure severity scale [31]. This is a 13-item scale covering areas of consciousness (4 questions), motor symptoms (2 questions), incontinence (1 question), injuries/pain (3 questions) and overall seizure severity (SS) (3). The tool has an internal consistency reliability (Cronbach's alpha = 0.85) and high test-retest reliability (Pearson correlation coefficient = 0.93). There is a point response scale and the sum of all items is the total scale score, (possible range: 13–53) where higher scores indicate greater seizure severity. It has been widely used in epilepsy-related research.

**Table 1. ADL, IADL and HDL with their descriptions.**

| Activity group | Specific activities |
|---|---|
| Activities of daily living (ADL) | Aiding patient with personal care<br>Aiding patient in visiting the toilet |
| | Aiding patient in moving around within the house<br>Aiding patient with eating and drinking |
| Instrumental activities of daily living (IADL) | Aiding patient in travelling outside the house<br>Aiding patient in making trips and visiting family or friends<br>Aiding patient in visiting a doctor or the hospital<br>Aiding patient in organizing help, physical aids or house adaptations<br>Aiding patient in taking care of financial matters like buying and selling things, etc. |
| Household activities of daily living (HDL) | Aiding patient in food and drink preparation<br>Aiding patient in washing the dishes<br>Aiding patient in cleaning the house or compound<br>Aiding patient in washing, mending, ironing clothes<br>Aiding patient in taking care of and playing with your children<br>Aiding patient in shopping household supplies (incl. food)<br>Aiding patient in other tasks like grinding and pounding, collection of water and fuel (e.g. firewood, charcoal) |

## Ethical statements

Ethics approvals were obtained from Ethics Committee's declaration of the School of Tropical Medicine and Global Health, Nagasaki University on January18th 2019 (number 2019–67) and Uganda National Council for Science and Technology (SS4555). Written informed consent were obtained from all participants in the study.

## Data analysis

Costs were assessed over a 1-month period, preceding the data collection. They were analyzed from the caregiver perspective. The direct costs were also adjusted for inflation for the year 2018. A summary of the direct (medical and non-medical) costs and indirect (informal care-giving) costs is presented in Table 2. However, intangible costs which constitute an essential part of cost burden for a health condition have not been considered in this study. Intangible costs are mainly the non-physical costs that a patient(s) and their families experience due to ill health or while undertaking treatment and are difficult to measure and assess, for instance the pain and suffering associated with the disease. They focus on caregivers' subjective burden, health and well-being assessed from their perspective and as have been stressed by Hoefman et al. [29], the measurement of subjective burden is considered a non-monetary valuation.

In order to compute the per-patient cost, in households with more than one patient, the direct and indirect costs were divided by the number of patients in the household. Direct costs were computed by adding the expenditure on item(s) under the direct medical costs, to the

**Table 2. Summary of study variables and their description.**

| Variable | Subcategory | Description |
|---|---|---|
| Direct costs | Direct medical | • Registration fees, consultation fees, cost of diagnosis (tests; laboratory and any other), cost of medications (drugs), hospitalization fees |
| | Direct non-medical | • Transportation, accommodation, meals |
| Indirect costs | | • Informal caregiving costs |

ones under direct non-medical costs and then weighted by the proportion of people who indicated having spent on such item(s) under each category. Informal care costs were assessed using the opportunity cost method of the human capital approach [32,33] in valuing unpaid labor such as household work and childcare among others. Broadly speaking, the opportunity cost method estimates (at market wage rate) the value of informal caregiver benefits forgone as a result of spending time on providing informal care [34]. Therefore, according to this method, the value of informal caregiving (VIC) = $\beta_i W_i$, where; $\beta_i$ is the number of hours spent on the informal caregiving task by the principal caregiver and $W_i$ is the hourly wage of the caregiver. In addition, employment effects (labor force participation and employment rates) were considered in accordance with literature on the human capital approach [35], as not all household members participate in economic activities [36].

Since no respondents were formally employed and it has been reported that men on average earn twice (134,042 Ugandan shillings; ~37.2 USD) as compared to women (67,021 Ugandan shillings; ~18.6 USD), a proxy was used in computing the informal care costs: the real national median monthly wage (gender-and age-specific) times the labor force participation rate (39.6%) and employment rate (90.6%) for the Acholi sub-region. All of these figures were extracted from the country's 2018 Statistical abstract [37]. The information for the abstract is based on "latest surveys, censuses and administrative records of Ministries, Departments and Agencies which are disseminated for use in tracking outcomes of policies as well as decision-making" [37]. Therefore, these figures can reflect the rural-urban work-related disparities.

Leisure time was not valued in the current study since it's considered a non-productive activity in accordance with Reid's third-person criterion: "If an activity is of such character that it might be delegated to a paid worker, then that activity shall be deemed productive" [38]. For instance, one cannot hire someone to eat, sleep or learn on one's behalf. This difficulty of delegation makes it hard to attach a monetary value to foregone leisure time. Drummond et al. acknowledges this but argues that valuation can be done from zero, through average earnings, to average overtime earnings [39]. Despite these suggestions, it has remained a contentious issue and as [40] puts it; '. . .*valuing the leisure time of an individual according to their wage rate may prove difficult and ethically debatable*.' Leisure time is thus, proposed to be valued in terms of quality of life [41].

Informal caregiving costs per patient were computed as follows to generate a monthly estimate: [weekly total reported foregone hours to provide informal care × 4 (average number of workweeks in a month) × real median hourly wage (gender-and age-specific) [42] x employment rate (for the Acholi sub-region) x labor force participation rate (for the Acholi sub-region)].

In order to generate the yearly estimates, the 4-average number of workweeks in a month was instead replaced by 52 average number of workweeks in a year.

These were then weighted by the proportion of people who indicated having foregone unpaid work to provide informal care. Direct and indirect costs were converted to USD at an exchange rate of 1 USD = 3704 Ugandan shillings. For meaningful comparison to studies from other countries, we inflated all local currencies of the study year to 2018 figures using the Consumer Price index (CPI) and then later converted to 2018 international dollars ($) using the power purchasing parity (PPP) for private consumption. All indicators used in these conversions were from the World Bank Official website [43].

## Results

### Background characteristics of caregivers and receivers

There were 14 caregivers recruited for this study and except for one male caregiver (Table 3), all caregivers were females (92.9%). The average caregiver age was 42.1years (range: 21–56).

**Table 3. Socio-demographic characteristics of caregivers.**

| Background Characteristic | Freq (No.) | Percent (%) | Mean (SD) | Range |
|---|---|---|---|---|
| **Sex** | | | | |
| Male | 1 | 7.1 | | |
| Female | 13 | 92.9 | | |
| **Age** | | | | |
| 18–34 years | 3 | 21.4 | | |
| 35–44 years | 6 | 42.9 | 42.1 (9.3) | 21–56 |
| 45–54 years | 4 | 28.6 | | |
| 55 years and above | 1 | 7.1 | | |
| **Marital status** | | | | |
| Married | 8 | 57.1 | | |
| Divorced/Separated/Widowed | 6 | 42.9 | | |
| **Educational qualification** | | | | |
| No education | 6 | 42.9 | | |
| Formal education | 8 | 57.1 | | |
| **Occupation** | | | | |
| Farmer | 14 | 100 | | |
| Formal employment | – | – | | |
| **Household size (including NS patients)** | | | | |
| ≤ 5 members | 2 | 14.3 | | |
| 6–10 members | 10 | 71.4 | 8.3 (2.3) | 4–13 |
| >10 members | 2 | 14.3 | | |
| **NS patients in a household** | | | | |
| 1 | 8 | 57.1 | | |
| 2 | 3 | 21.4 | 1.7 (1.0) | 1–4 |
| 3 | 2 | 14.3 | | |
| 4 | 1 | 7.1 | | |
| **Caregiver monthly reported income (In USD)** | | | | |
| 0–20 | 2 | 14.3 | | |
| 21–30 | 5 | 35.7 | 31.6 (11.1) | 13.2–55.8 |
| 31–40 | 4 | 28.6 | | |
| 41–50 | 2 | 14.3 | | |
| 51+ | 1 | 7.1 | | |
| **Perceived financial status** | | | | |
| Well-off | – | – | | |
| Fairly well-off | – | – | | |
| Average | 8 | 57.1 | | |
| Fairly poor | 5 | 35.7 | | |
| Poor | 1 | 7.1 | | |
| **Total** | 14 | 100.0 | | |

SD: standard deviation

More than half (57.1%) of the respondents were married and stopped their education at either some level in primary school or after completing primary school. All respondents were farmers. Some respondents also engaged in small businesses as a secondary means of livelihood. The reported mean monthly income of the caregivers was 31.6 USD (range: 13.2–55.8), however, half (50%) of the households reported income below the average. Nearly three-quarters

of the households (71.4%) had 6–10 members (range: 4–13). On average, there were about 2 NS patients per household.

There were 24 NS patients who participated in the study (14 males; 58.3% and 10 females; 41.7%) (Table 4). The median age of all patients was 20 years (Inter Quartile Range:17.5–21.5).

**Table 4. Socio-demographic characteristics of NS patients.**

| Background characteristic | Freq (No.) | Percent (%) | Mean (SD) | Median (IQR) | Min | Max |
|---|---|---|---|---|---|---|
| **Sex** | | | | | | |
| Male | 14 | 58.3 | | | | |
| Female | 10 | 41.7 | | | | |
| **Age** | | | | | | |
| ≤ 17 years | 6 | 25.0 | 19.2 (4.7) | 20.0 (17.8–21.3) | 6 | 27 |
| ≥ 18 years | 18 | 75.0 | | | | |
| **Marital status** | | | | | | |
| Never married | 17 | 70.8 | | | | |
| Married /Divorced /Separated | 7 | 29.2 | | | | |
| **Educational qualification** | | | | | | |
| No education | 8 | 33.3 | | | | |
| Some / complete primary | 16 | 66.7 | | | | |
| **Relationship to caregiver** | | | | | | |
| Son/daughter | 18 | 75.0 | | | | |
| Others | 6 | 25.0 | | | | |
| **Residence Status** | | | | | | |
| Co-resident with caregiver | 19 | 79.2 | | | | |
| Nearby caregiver | 5 | 20.8 | | | | |
| **Reported duration of being with illness**. | | | | | | |
| (1–10) years | 9 | 37.5 | | | | |
| 11 years and above | 15 | 62.5 | | | | |
| **Can patient be left alone?** | | | | | | |
| No, needs constant surveillance | 1 | 4.2 | | | | |
| Yes, not more than 1 hr | 3 | 12.5 | | | | |
| Yes, for several hours | 20 | 83.3 | | | | |
| **Last seizure experience** | | | | | | |
| Up to 1 week ago | 18 | 75.0 | | | | |
| Up to 1 month ago | 5 | 20.8 | | | | |
| 3–6 months ago | 1 | 4.2 | | | | |
| **Seizure frequency** | | | | | | |
| Daily /multiple per day | 5 | 20.8 | | | | |
| Weekly but not daily | 16 | 66.7 | | | | |
| Monthly but not weekly | 3 | 12.5 | | | | |
| **Seizure Severity** | | | | | | |
| Very severe | 9 | 37.5 | | | | |
| Severe | 9 | 37.5 | | | | |
| Mild and very mild | 6 | 25.0 | | | | |
| **Hague Seizure Severity Scale score** | – | – | 25.1 (6.8) | 22.5 (21.0–27.5) | 16 | 41 |
| **Total** | 24 | 100.0 | | | | |

SD: standard deviation

IQR: interquartile range

More than half, 70.8% were never married, 66.7% either stopped their education at some level in primary school or after completing primary school and 79.2% of the patients were sons /daughters of caregivers. More than three-quarters (79.2%) of the patients experienced weekly or monthly, but not daily seizures, with about 1 in 4 patients experiencing mild seizures. Seizure severity was assessed using the Hague seizure severity scale with a mean (standard deviation (SD)) score of 25.1 (6.8).

### Direct cost of NS

The monthly weighted mean direct cost per NS patient was 2.2 USD (with a weighted median cost of USD 1.3) corresponding to an annual cost of 26.4 USD (Table 6). Direct cost accounted for about 7.7% of household expenditure, with more than half of the cost incurred on travel and cumulatively, over 90% primarily incurred on direct non-medical costs.

### Indirect cost of NS

The indirect cost was caregivers' valued (unpaid care) time devoted to taking care of NS patients. This time accounted for less than half (~ 40%) of all forgone care time as leisure time was not valued. Therefore, the average ± SD (standard deviation) caregiving time per week was estimated at 2.4 (± 0.7) hours with a total of 14.5 hours (Table 5). The monthly weighted mean indirect cost per NS patient was 0.1 USD (and zero weighted median cost) (Table 6).

### Total cost of NS

The weighted mean monthly sum of direct and indirect costs per NS patient was estimated at 2.3 USD (96.2% direct cost, 3.8% indirect cost) (Table 6), corresponding to an annual cost 27.6 USD.

The total weighted median monthly cost has been estimated at 1.4 USD.

## Discussion

The study results show that direct costs (out-of-pocket expenses) were a major driver of NS costs, constituting a significant portion of household expenditure and accounted for 96.2% of all costs. The larger proportion of the direct costs were mainly non-medical costs. These direct costs could have been even higher if the following were not in place: (i) the free provision of treatment by the (Ugandan) government; and, ii) transport support provided to this group of patients by the local community-based organization operating in the area. This was in line

**Table 5. Informal care time in mean hours per caregiver per week.**

| | No. (%) of caregivers who reported foregoing paid, unpaid or leisure time in order to provide informal care per week | | Total hours spent on informal caregiving per week | Mean (standard deviation) hours spent informal caregiving per week | |
|---|---|---|---|---|---|
| Foregone paid work time to provide informal care | – | – | – | – | – |
| Foregone unpaid work time to provide informal care | 6 | (42.9) | 14.5 | 2.4 | (0.7) |
| Foregone leisure time to provide informal care | 11 | (78.6) | 21.5 | 2.0 | (1.0) |
| Total | 14 | – | 36.0 | 4.4 | (1.7) |

SD: standard deviation

**Table 6. Direct and indirect (informal caregiving) costs of NS.**

| Cost category | No. (%) caregivers who incurred direct and indirect costs | Mean (SD) Monthly amount (USD) | Weighted mean monthly amount (US$) | Median (IQR) monthly amount (US$) | Cost profile (%) |
|---|---|---|---|---|---|
| **DIRECT COST** | | | | | |
| Medical[a] | | | | | |
| Booklet purchase | 13 (92.9) | 0.1 (0.0) | 0.1 | 0.0 (0.0–0.1) | 2.4 |
| Non-medical[b] | | | | | |
| Meals | 11 (78.6) | 1.1 (2.3) | 0.8 | 0.3 (0.3–1.0) | 35.9 |
| Travel[c] | 14 (100.0) | 1.3 (0.8) | 1.3 | 1.1 (0.5–1.6) | 57.9 |
| **Total Direct Cost** | – – | **2.5 (3.1)** | **2.2** | **1.4 (0.8–2.6)** | **96.2[d]** |
| **INDIRECT COST[e]** | | | | | |
| Valued foregone time devoted to informal caregiving | 6 (42.9) | 0.2 (0.1) | 0.1 | 0.1(0.1–0.2) | 3.8 |
| **Total Indirect Cost** | – – | **0.2 (0.1)** | **0.1** | **0.1(0.1–0.2)** | **3.8** |
| **Total costs*** | – – | **2.7 (3.2)** | **2.3** | **1.5(0.9–2.8)** | **100.0** |

Weighted median direct cost was USD 1.3 (0.929*0.0 + 0.786*0.3 + 1*1.1) and indirect costs was USD 0.0 (0.429*0.1).

1 USD = 3,704 Ugandan shillings; SD: standard deviation; IQR: interquartile range and there were 24 patients.

[a] Treatment is being provided free by the government of Uganda.

[b] Only 2 households incurred accommodation expenditure; (one incurred accommodation costs worth 75 USD, more than all direct costs combined (53.55 USD) and the other 4.2 USD, giving an average of 39.6 USD for these two households).

[c] Travel costs are being supported by a local community-based organization in the area.

[d] Direct costs account for 7.7% of household expenditure (excluding the amount for accommodation).

[e] Only foregone unpaid work time (approximately (~) 40% of all foregone time) was valued, leisure time was not valued.

[f] Direct and indirect costs were computed per NS patient

with a study by [44] which showed that the lack of money for transport and medical bills was a big challenge in accessing health care for NS patients and their households.

In comparison to other studies from sub-Saharan Africa (when all costs of all studies were converted to 2018 international dollar ($) figures) (Table 7), the annual average direct cost of NS per patient for the current study was estimated at 75.5 USD. This was more than that reported among epilepsy patients in Burundi (43.6 USD) [45] where a significant amount (6.13 USD) was spent on anti-epileptic treatments alone. In the current study, such treatment is provided free of charge to the NS patients. However, Nsengiyumva et al. did not address direct non-medical costs in their study, which could have potentially led to a rise in their direct costs if included [45]. A study from rural South Africa with some similar conditions to the current study, including the community-based free provision of anti-epileptic drugs, estimated the annual average direct cost of epilepsy per person at a clinic or hospital at 125.0 USD [46]. This was more than the direct costs reported in the current study (75.5 USD). Wagner et al. also reported that more than half (67%) of their study participants sought epilepsy care from both a biomedical facility and a traditional healer, with some individuals seeking care from only traditional healers [46].

Similarly, Siewe Fodjo et al. estimated the annual average direct cost per person with epilepsy at 366.0 USD in the Democratic Republic of Congo [47]. Again, this is more than four times the direct cost reported in the current study. It is worth noting that Siewe Fodjo et al. found that anti-epileptic drugs (19.8%) and traditional medicine (68.2%) accounted for more than three-quarters (88%) of all direct costs, and were thus, the greatest drivers of direct costs in their study [47]. From these studies, it can be seen that traditional medicine is an integral

Table 7. Annual cost of epilepsy from some African countries.

| Country /Year of study | Study setting and methods | Annual direct cost (I$[a]) | Annual indirect cost (I$[a]) | Annual total cost (I$[a]) |
|---|---|---|---|---|
| Burundi, 1998 [45] | 1. Case-control study evaluating cost of care for PWE vs persons without epilepsy<br>2. Recruitment in a private hospital<br>3. Direct cost: consultation, admission and complementary exams, AED cost<br>4. Indirect cost: number of days of family life disrupted expressed as a multiple of the GDP per capita per day | 43.6 | 222.8 | 266.3 |
| Nigeria, 2012 [51] | 1. Cross-sectional, case study, conducted at a tertiary health facility<br>2. Recruitment of study subjects done at the facility<br>3. Direct costs: Transportation, consultation/registration, laboratory investigations drug, others<br>4. Indirect costs: Number of days of work lost by caregiver, expressed as a factor of daily income loss | 403.1 | 204.0 | 607.1 |
| South Africa, 2011 [46] | 1. Community-based study with home visits to previously diagnosed PWE<br>2. Healthcare utilization and out-of-pocket expenditure for epilepsy were evaluated<br>3. Direct cost: visit to clinic/hospital, transportation, food/drinks purchased due to visit, traditional healer cost<br>4. AEDs freely available for PWE, thus no AED cost included<br>5. Indirect cost was not evaluated | 125.0 | NA | 125.0 |
| Democratic Republic of Congo, 2017 [47] | 1. Cross-sectional study evaluating cost of care of PWE on families<br>2. Recruitment in government health centres<br>3. Direct cost: consultation, hospitalization, complementary exams, AED cost, traditional medicine cost, and transportation<br>4. Indirect cost: number of days of work lost by PWE and caretaker, expressed as a multiple of the GDP per capita per day | 366.0 | 362.4 | 728.5 |
| Uganda, 2019 (Current study) | 1. Community-based cross-sectional cost-of-care study for NS households<br>2. Recruitment of study participants done at the community<br>3. Direct costs: Registration, transportation, accommodation, meals<br>4. Indirect costs: foregone time to provide care for NS patients, expressed in terms of gender-and age-specific hourly income loss | 75.5 | 3.4 | 79.0 |

[a]I$: 2018 International dollars.

part of the costs for epilepsy patients [46,47]. Atim et al. had earlier reported that caregivers of NS patients did not only seek care from the health facility but also went to traditional healers or/and faith-based healers [48]. However, in the current study, use of traditional medicine by NS patients was not assessed. The failure to assess the use of traditional medicine may have underestimated the direct costs reported in the current study. At the same time, it can also be argued that most (if not all) of the patients in this study population may actually be getting their medication from the health centers (due to the free treatment provision and transport support). As the researcher was a resident in this community for a period of about two months and in informal conversations with caregivers, community members reported that the condition of patients who took the medications regularly improved as they witnessed a reduction in seizures (according to their understanding and observation) which could also support the idea of seeking care from the health centers. However, those patients who did not take the medications (as they were said to be allergic to the drugs; according to the understanding of the caregivers) were reported to have not improved (as they had frequent seizures even at times daily). Care seeking means (including alternative care-seeking) for patients said to be allergic to the

(anti-epileptic) medication was not assessed unfortunately, and thus, future research could focus on these two areas identified.

The total indirect cost, based on the foregone time to provide informal care to NS patients, which time was estimated at 36 hours per week (14.5 hours on forgone unpaid work and 21.5 hours on foregone leisure time), averaging 4.4 ± 1.7 hours per week. This number of hours could be considered relatively small. However, it has been well noted that care time reported by parents often looks very short compared to the perceived burden, due to in part; the definition of care time itself [49]. This is especially the case when care time is absorbed, most often unconsciously, into domestic work, and thus caregivers end up only reporting their time involvement in "active" care while neglecting "passive" care time in terms of watching over or keeping an eye on their children. Accordingly, caregivers' inability to report passive care time greatly under estimates caregiving. Moreover, studies have shown that most NS patients have seizures mainly in the morning [5]. While studying time allocation of caregivers (as part of the current study, but not reported here), the researcher was at times able to witness these early-morning seizures. For a community afflicted with NS for a relatively long time, it can be argued that they might have adjusted in a way to deal with the condition. The occurrence of seizures at such (morning) times suggests that available family members (who could be engaged in whatever activity at that time but within the vicinity where such a patient might be having seizures) would readily come to the aid of the patient(s) without necessarily involving or needing the primary caregiver. Thus, a patient(s)' care would generally be distributed among members of the household, extended families, and sometimes neighbors. This is evident due to the existence of familial ties between some of the households. Some of the patients were not stationary in their main homes; they could move to their relatives' homes and spend days (or even weeks) there. The care for such patients was therefore not totally dependent on the main caregiver, due to their constant movements and this might explain why the few hours of care were reported. Other studies assessing informal caregiving burden have conversely reported considerably higher hours of informal care involved. For instance, in assessing the family caregiving burden for the elderly in southern Ghana, Nortey et al. reported average caregiving hours of 54.9 (± 6.5) per caregiver per week [50]. However, they stated that since co-resident caregivers might have found it difficult to distinguish time spent in care tasks from other household duties, care time could have been over-reported in their study. That said, the informal caregiving hours reported in the current study could be considered relatively low in comparison.

The mean annual informal care costs for this group of NS patients was estimated at 3.4 USD (Table 7). In comparison to studies from sub-Saharan Africa, Nsengiyumva et al. reported indirect costs of epilepsy of 222.8 USD [45], Siewe Fodjo et al. reported costs of 362.4 USD [47] and Ughasoro reported costs of 204.0 USD [51], although, it's worth noting that Ughasoro et al. conducted their study at a teaching hospital [51]. All three of these sub-Saharan studies reported far greater indirect costs than the current study. Several possible scenarios could explain the low informal care costs. The first plausible explanation is the relatively few hours of informal care reported (4.4 hours with 2.4 on forgone unpaid work and 2.0 on forgone leisure time) per week, of which, only the forgone time on unpaid work (2.4 hours) was valued. Leisure time (which accounted for about 60% of all forgone time, Table 5) was not valued, which could partly explain the low informal care costs reported in this study. Secondly, this study is a case series with a small sample size; (only 14 caregivers' costs were evaluated, patient costs were not evaluated in this study), while the other studies had much larger samples sizes; for instance, Nsengiyumva et al. had more than 1000 participants [45], while Siewe Fodjo et al. had more than 250. Thirdly, the different valuation methods used could account for the variations as well. For instance, Nsengiyumva et al. considered days of family life

disrupted, [45] while Siewe Fodjo et al. considered number of days of work lost by epilepsy patients and caregiver instead of the actual hours foregone [47]. In terms of the indirect cost estimation, the studies above studies valued informal care by simply multiplying the number of disrupted or lost days by the GDP per capita per day. However in the current study, a proxy wage, the real national median monthly wage (gender-and age-specific) was used to value fore-gone time. The GDP per capita per day was instead used to estimate the would-be loss of GPD to the nation. This approach has been used by Brinda et al. in valuing the cost and burden of informal caregiving of dependent older people in a rural Indian community [52]. The valua-tion method (opportunity cost method) could also account for the low indirect cost, especially because of its inclusion of employment effects, such as the labor force participation rate, which has been reported to actually undervalue earnings [35] as has been demonstrated by Hanly et al. [53]. In using the opportunity cost method to value informal caregiving in Ireland, Hanly et al. considered three different scenarios [53]. In the first scenario (Scenario1), they applied gender-and age-specific national median wages to value informal caregiving time which became a base estimate. In Scenario2, they incorporated Scenario1 with caregiver labor market employment effects (labor force participation and employment rates) but did not value infor-mal caregivers outside of paid employment, (in other words, unpaid caregivers were excluded). In Scenario3, they incorporated Scenario1 with Scenario2, but also included unemployed care-givers and valued informal caregiving using the minimum wage. In Scenario3, which is most similar to the case in the current study (as both include unpaid caregivers), they found that the application of the employments effects (labor force participation and employment rates) led to a 42% (for the male valued informal care costs) and 62% (for the female valued informal care costs) reduction in the total value of informal care cost below the base estimate. Thus, the opportunity cost method can be said to undervalue informal care costs. Broadly speaking, the indirect costs estimated for the current study can be considered low in comparison to the other discussed studies from sub-Saharan Africa. In terms of the would-be loss of GPD to the nation as a result of informal careging; the Gross Domestic Product (GDP) of Uganda in 2018 was estimated at 27.477 billion US dollars (USD) for a total population of 42,723,139 persons [54]. This equates to a GDP per capita of 643.14 USD, GDP per capita per day of 1.76 USD, and the GDP per capita per hour of 0.2203 USD. Therefore, the would-be GDP contribution to the nation if all forgone time (for both unpaid work and leisure) spent by these caregivers in a year were to be added to the productive labor working hours, the GDP of the nation could increase by 412.40 USD [GDP per capita per day per hour (0.2203 USD) x 1872 (total foregone hours in a year)]. The caregivers of the NS patients in the present study were predominantly women. As found out, majority of the fathers of NS patients normally distance themselves from the care of such children, thus letting the care responsibility solemnly fall on the women [55]. The Hague severity scale scores ranged from 16–41, with a mean ± SD of 25.1 ± 6.8. The median score was 22.5, with an interquartile range of 21.0–27.5. The scale scores range from 13–53, where higher scores normally indicate greater seizure severity. The distribution of patients' severity scores in the current study with the range of Hague severity scale range (13–53) suggests that, overall, the majority of the patients experienced average seizure severity. It is however, worth noting that, this scale measures severity based on the parent's perception of the child's seizures and it is thus a subjective assessment. Additionally, only a handful of the patients had daily seizures. This could be linked to the provision of free treatment and access to facilitated transportation to health facilities, which enabled patients to receive medications promptly. Thus, most patients' condition could be said to have improved, as most of the patients had only either weekly or monthly seizures. In fact, most patients were able to do sports (playing football) or fetch water in the company of other family members. However, as previously stated, due to the nature of the setting, where there exists strong familial ties, it is

possible that the direct and indirect costs were not entirely borne by the primary caregiver alone; other family members might have contributed in caring and supporting with the direct costs for these patients. In other words, the burden may have been shared. Surprisingly, a majority of households perceived their financial situation as average. This could be linked to the low hours (4.4 hours per week) of informal caregiving reported in the current study. According to Thomson et al. (as cited in [50]), caring for 20 hours or more per week among family caregivers was associated with a higher risk of financial stress among family caregivers [50]. This could be attributed to caregivers having less time to participate in the labor market, consequently reducing their income and leading to financial stress. Nakigudde et al. revealed that, in the absence of healthcare services or support groups that would help in alleviating the fulltime caregiving burden on NS households, caregivers remain vulnerable and become financially constrained due to the interference in their day-to-day (agricultural production) activities [22]. This is because such households have been known to mainly rely on subsistence farming and would primarily generate their family income by selling farm produce. The reliance on subsistence agriculture alone could also be a reason that the majority of such households perceived their financial status as average.

This study, however, acknowledges the following limitations. First, the study did not assess whether patients also accessed traditional medicine, as some studies have shown that expenditure on traditional medicine could constitute an integral part of direct costs for epilepsy patients [46,47]. In addition, patient productivity loss was not assessed in this study, therefore, assessing economic loss to households and society can be a further area of research, especially since the majority of the patients themselves are now at a productive age. Since the primary caregivers had not obviously spent their whole time in caring for these patients, secondary caregivers such as other members of the extended family as well as neighbors might have contributed to this activity to a certain degree. Therefore, we therefore acknowledge that the failure to assess the indirect and even direct cost burden sharing by other family members and neighbors in our study would no doubt result to underestimation of the cost. Furthermore, since this study was done in the community, it could be possible that some epilepsy patients might have been regarded as NS patients since there is no clear community distinction between the two (NS and epilepsy). However, by following the MoH summary statistics, which showed that no new cases have been reported since 2012 [15], it can be presumed that the NS patients may be known in their communities. Another limitation of our research is that it is a cross-sectional study with a small sample size and is thus not generalizable to all NS households. Only descriptive statistics were presented, no statistical tests were performed. A larger case-control study would be needed to further confirm these relations. There could also be a possibility of recall bias, as respondents were asked to estimate their average time foregone or spent on certain activities; therefore, it was arguably difficult to report accurate time foregone.

## Conclusion

This study sought to examine the direct and indirect costs of NS on the caregivers so as to inform public health interventions that could help lessen the burden in order to improve the quality of life for NS households and ultimately the nation at large. Overall, the study found that; direct costs constituted a higher proportion of costs for NS households, accounting for, on average, 7.7% of household expenditure. Therefore, more studies are needed to investigate measures that could help bring down these costs. For instance, encouraging mechanisms that support affected households (or patients in particular) in accessing anti-epileptic treatments in a timely manner such as stocking anti-epileptic drugs with village health teams who could distribute them or simply allow households to pick from them, thus significantly reducing the

direct costs and disruption in day-to-day activities of caregivers. Alternatively, in going to collect patients' medications from the health facility, patients' conditions could be assessed and those said to be allergic (according to the understanding of caregivers) could potentially be given alternative medication.

## Supporting information

**S1 File.**
(DOCX)

## Author Contributions

**Conceptualization:** Lugala Samson Yoane Latio, Nguyen Hai Nam, Nishi Makoto.

**Data curation:** Lugala Samson Yoane Latio, Jaffer Shah.

**Formal analysis:** Nguyen Hai Nam, Jaffer Shah, Kato Stonewall Shaban, Richard Idro.

**Investigation:** Lugala Samson Yoane Latio, Nguyen Hai Nam, Kikuko Sakai, Richard Idro.

**Methodology:** Nguyen Hai Nam, Jaffer Shah, Nishi Makoto.

**Supervision:** Chris Smith, Nguyen Tien Huy, Shinjiro Hamano, Kazuhiko Moji.

**Validation:** Jaffer Shah, Chris Smith, Kato Stonewall Shaban, Nishi Makoto.

**Visualization:** Nguyen Hai Nam, Jaffer Shah, Kikuko Sakai, Richard Idro.

**Writing – original draft:** Lugala Samson Yoane Latio, Nguyen Hai Nam, Kikuko Sakai.

**Writing – review & editing:** Chris Smith, Kato Stonewall Shaban, Richard Idro, Nishi Makoto, Nguyen Tien Huy, Shinjiro Hamano, Kazuhiko Moji.

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
