## [Decision Letter · Decision Letter 0]

19 May 2020

PONE-D-20-04251

Economic burden of the persistent morbidity of nodding syndrome on affected households in Northern Uganda.

PLOS ONE

Dear Dr. Huy,

Thank you for submitting your manuscript to PLOS ONE. After careful consideration, we feel that it has merit but does not fully meet PLOS ONE’s publication criteria as it currently stands. Therefore, we invite you to submit a revised version of the manuscript that addresses the points raised during the review process.

We would appreciate receiving your revised manuscript by Jul 03 2020 11:59PM. To enhance the reproducibility of your results, we recommend that if applicable you deposit your laboratory protocols in protocols.io, where a protocol can be assigned its own identifier (DOI) such that it can be cited independently in the future. For instructions see: http://journals.plos.org/plosone/s/submission-guidelines#loc-laboratory-protocols

We look forward to receiving your revised manuscript.

Kind regards,

Xiangming Fang

Academic Editor

PLOS ONE

2. During our internal checks, the in-house editorial staff noted that you conducted research or obtained samples in another country. Please check the relevant national regulations and laws applying to foreign researchers and state whether you obtained the required permits and approvals. Please address this in your ethics statement in both the manuscript and submission information.

4. We note you have included a table to which you do not refer in the text of your manuscript. Please ensure that you refer to Table 1 in your text; if accepted, production will need this reference to link the reader to the Table.

Reviewers' comments:

Reviewer's Responses to Questions

**Comments to the Author**

1. Is the manuscript technically sound, and do the data support the conclusions?

Reviewer #1: Yes

Reviewer #2: Partly

2. Has the statistical analysis been performed appropriately and rigorously? 

Reviewer #1: Yes

Reviewer #2: Yes

3. Have the authors made all data underlying the findings in their manuscript fully available?

Reviewer #1: Yes

Reviewer #2: No

4. Is the manuscript presented in an intelligible fashion and written in standard English?

Reviewer #1: Yes

Reviewer #2: Yes

5. Review Comments to the Author

Reviewer #1: This paper examined the economic burden of nodding syndrome, a rare fatal neurological condition with unknown etiology. It fills gap in the literature but raises a few questions as well. I have some comments for the authors to consider during their revision of the manuscript.

First, the authors indicated that the patients’ productivity loss is not counted because the majority of the patients are not in the productive age. I doubt this is true as the summary statistics suggest that 75% of the patients are over age 18 years old. At the very least, the authors could examine the other economic evaluations that have provided much higher estimates to see if they have accounted for the patients’ productivity loss and the scale of that component.

Second, indeed the sample size is unfortunately very small so this is essentially a pilot study. To be fair to the authors, they have acknowledged the limitation. They might want to caution against the generalization of their results. I would expect more details of their sample selection. I took a look at the prevalence of nodding syndrome and do not think the 19 households represents all patients’ families in the district – what criteria the authors have used in identifying the households?

The authors also mentioned that the caregivers’ hours may be low because patients could be in extended families or neighbors. While this is true, I think this also lead to an omission of the cost (hours lost of the extended family members or neighbors) that should be accounted.

Specific comments

Page 6, line 86, “effective life-quality campaign”, please be specific.

Page 6, line 86, low-income country: country should be plural.

Line 96, remove extra space after “[11, 14]”

Line 101, “effect” does not seem to be the best choice of word, consider revising.

Line 111, from the context, low household income is not a risk factor but an associated factor.

Line 125, remove extra space before “with about 323..”

Line 127, add “patient” after “NS”

Line 133-135, the ethics statement is duplicative – move and merge with the later section on ethics approval.

Line 142-151, I suggest listing these descriptions in a table.

Line 222, the link to the website is not necessary as it is in the citation.

Line 223, replace “American” with “US”

Line 237, does “single” here mean “single in marital status” or just one person? Maybe rephrase to avoid confusion.

Line 240, peasant and farmers are repetitive

Reviewer #2: This paper examines the economic burden of nodding syndrome on affected households in Northern Uganda using a cross-sectional survey data. This paper studies an interesting topic that may have important policy implications. It has strength such as having a unique data and conducting thorough literature review. However, it presented major problems that prevented me from recommending publication. My major concerns and comments are the following:

• Title

o The title is a little misleading as the paper actually examines the economic burden of NS on the caregiver. Household also includes the patient but the cost doesn’t not consider any cost directly related to NS patients.

• Introduction

o The authors didn’t demonstrate a strong motivation of this paper. Line 104 to 111 discussed a little bit but it is not strong enough. Any serious child disease, which requires constant-caregiving, could fit in the context the authors discussed. Why looking at economic burden of NS is important? How the cost estimate information could be used for policy makers?

• Methods

o Study population

• The study was conducted from 1/2019 to 2/2019. When did the direct cost occur? Were the direct costs adjusted for inflation? If so, which year were they adjusted for?

• 14 caregivers and 24 participants were in the sample. How were the participants selected? Was the selection random? What is the selection criteria?

o Data analysis.

• Costs were analyzed from the caregiver perspective. Did you assume that caregiver pay for the costs? Is it possible that caregiver only provides care but not pay for the cost? If so, the wording needs to be revised.

• The authors should provide more details regarding how the direct costs were calculated.

• Line 200, should be “times” not “plus”

• The wage rate used to calculate the indirect costs is the real median monthly wage. Does this wage take into account age and sex factor? If not, in this sample, the majority are women, using a generic wage rate would not be appropriate if there is a substantial gender wage gap. The authors may consider using a weighted average wage based on the gender proportion in the sample.

• The discussion of the relation of informal caregiving cost to GDP could be moved to the discussion as it is not part of the cost calculation. (line 222 to 228)

• The authors didn’t provide a monetized value of intangible cost but instead provided caregiving burden. I wonder if the authors may find a way to obtain measures similar to the monetized QALY to capture the intangible costs? If not, it doesn’t make a lot of sense to show it along with direct cost and indirect cost.

• Line 233, was the alpha calculated based on the current data? If so, the sample size may be too small to be trusted.

o Results

• I am wondering how severe of the NS patients are in this sample? Is it possible for authors to present the average seizure severity score of this country patients? It is possible that patients with different severe level of NS may require different level of care which may affect the direct costs.

• The sample size is very small which may cause inaccuracy of the estimates of direct costs and time of spending on informal care giving.

• Line 254, this subtitle is confusing. Consider changing it to indirect cost.

• Line 270-279, there is no monetized value for the intangible cost, and thus not comparable to other costs. Considering removing this part.

o Tables

• Table 5 doesn’t seem to be directly relevant to the cost calculation.

6. PLOS authors have the option to publish the peer review history of their article (what does this mean?). If published, this will include your full peer review and any attached files.

Reviewer #1: No

Reviewer #2: No

---

## [Author Response · Author response to Decision Letter 0]

18 Jun 2020

June 19th, 2020

PONE-D-20-04251

Economic burden of the persistent morbidity of nodding syndrome on affected households in Northern Uganda.

Dear Editors and Reviewers,

PLOS ONE

Thank you for giving us the opportunity to submit a revised draft of my manuscript entitled “Economic burden of the persistent morbidity of nodding syndrome on caregivers in affected households in Northern Uganda” to PLOS ONE. 

We appreciate the time and effort that the Editors and the Reviewers have dedicated to providing your valuable feedback on our manuscript. We are grateful to the reviewers for their insightful comments on our paper. We have been able to incorporate changes to reflect most of the suggestions provided by the reviewers and we hope that our explanations and revisions will be deemed satisfactory. 

Best regards,

Associate Professor Nguyen Tien Huy, M.D., Ph.D.

Department of Clinical Product Development

Institute of Tropical Medicine - Nagasaki University.

******

Journal requirement #1: Please ensure that your manuscript meets PLOS ONE's style requirements, including those for file naming. The PLOS ONE style templates can be found at http://www.plosone.org/attachments/PLOSOne_formatting_sample_main_body.pdf and http://www.plosone.org/attachments/PLOSOne_formatting_sample_title_authors_affiliations.pdf

Author Response: Dear Editor, we have adjusted our manuscript that meets PLOS ONE’s style requirements.

Journal requirement #2: During our internal checks, the in-house editorial staff noted that you conducted research or obtained samples in another country. Please check the relevant national regulations and laws applying to foreign researchers and state whether you obtained the required permits and approvals. Please address this in your ethics statement in both the manuscript and submission information.

Author Response: Dear Editor, based on your comments, we clearly state the permission and approval in the ethics statement in both the manuscript and submission information as follows:

“Ethical statements

Ethics approvals were obtained from Ethics Committee’s declaration of the School of Tropical Medicine and Global Health, Nagasaki University on January18th 2019 (number 2019-67) and Uganda National Council for Science and Technology (SS4555). Written informed consent were obtained from all participants in the study.” (Page 9, line 282-286)

Journal requirement #3: Please include additional information regarding the survey or questionnaire used in the study and ensure that you have provided sufficient details that others could replicate the analyses. For instance, if you developed a questionnaire as part of this study and it is not under a copyright more restrictive than CC-BY, please include a copy, in both the original language and English, as Supporting Information.

Author Response: Dear Editor, we include an additional information regarding the questionnaire used in our study with sufficient details that others could replicate the analyses. Please find the the file “Research questionnaire (Supporting information).docx” for more detail.

Journal requirement #4: We note you have included a table to which you do not refer in the text of your manuscript. Please ensure that you refer to Table 1 in your text; if accepted, production will need this reference to link the reader to the Table.

Author Response: Dear Editor, we have adjusted our mistake by refering the mentioned table above in the text of our manuscript.

****** 

Reviewer #1: This paper examined the economic burden of nodding syndrome, a rare fatal neurological condition with unknown etiology. It fills gap in the literature but raises a few questions as well. I have some comments for the authors to consider during their revision of the manuscript.

Comment 1) First, the authors indicated that the patients’ productivity loss is not counted because the majority of the patients are not in the productive age. I doubt this is true as the summary statistics suggest that 75% of the patients are over age 18 years old. At the very least, the authors could examine the other economic evaluations that have provided much higher estimates to see if they have accounted for the patients’ productivity loss and the scale of that component.

Author Response: We are thankful to the Reviewer’s comment and totally agree to this remark. Assessment of the patient’s productivity loss is necessary and important. However, in our study, we mainly focused on evaluating the economic burden of the caregiver. The reason is that most of the patients had been reported with cognitive impairment. This condition would arguably prevent or interrupt them from working and therefore force caregivers to look after them. Given this, the failure of assessment of patient costs has been acknowledged as one of the limitations (Page 21, line 641-643) and we have also rephrased the study title to “Economic burden of the persistent morbidity of nodding syndrome on caregivers in affected households in Northern Uganda”. 

Text Insertion (if applicable)/ page/ line number of change:

(Page 1, line 1-2) “Economic burden of the persistent morbidity of nodding syndrome on caregivers in affected households in Northern Uganda.”

Comment 2) Second, indeed the sample size is unfortunately very small so this is essentially a pilot study. To be fair to the authors, they have acknowledged the limitation. They might want to caution against the generalization of their results. I would expect more details of their sample selection. I took a look at the prevalence of nodding syndrome and do not think the 19 households represents all patients’ families in the district – what criteria the authors have used in identifying the households?

Author Response: Dear Reviewer, we appreciate your valuable comments for the sample size. It is small and our study is considered as a pilot study. Regarding the criteria that we have used in identifying the households, this study was performed in a village located in Northern Uganda (Lekwala). The inclusion criteria consisted of the households with nodding syndrome patients, who are members of a community-based organization called Alliance for Community with Nodding Syndrome (ACNS), which is being supported by Uganda-Japan Nodding Syndrome Network. After screening, we identified only 19 households who meet the inclusion criteria in this village and among them, five were excluded due to translocation to other districts. There were finally 14 households who enrolled to our study. An additional part has been added into the manuscript to clarify the inclusion criteria. 

Text Insertion (if applicable)/ page/ line number of change:

(Page 8, line 217-223)

“..The inclusion criteria consisted of the households with nodding syndrome patients, who are members of a community-based organization called Alliance for Community with Nodding Syndrome (ACNS), which is being supported by Uganda-Japan Nodding Syndrome Network. After screening, we identified only 19 households who meet the inclusion criteria in this village and among them, five were excluded due to translocation to other districts. There were finally 14 households who enrolled to our study.”

Comment 3) The authors also mentioned that the caregivers’ hours may be low because patients could be in extended families or neighbors. While this is true, I think this also lead to an omission of the cost (hours lost of the extended family members or neighbors) that should be accounted

Author Response: Thank you again for your pointing out this issue. At first, we would like to state that our study mainly focused on primary caregivers. However, since they had not obviously spent their whole time in caring these patients, we acknowledged that secondary caregivers such as other members of the extended family as well as neighbors might have contributed to this activity to a certain degree. Given this, the exclusion of secondary caregivers would no doubt result to underestimation of the cost. Therefore, we would like to mention this limitation in the following part: 

Text Insertion (if applicable)/ page/ line number of change:

(Page 20, line 745-750)

“..Since the primary caregivers had not obviously spent their whole time in caring for these patients, secondary caregivers such as other members of the extended family as well as neighbors might have contributed to this activity to a certain degree. Therefore, we therefore acknowledge that the failure to assess the indirect and even direct cost burden sharing by other family members and neighbors in our study would no doubt result to underestimation of the cost..”

Comment 4) Specific comments

Page 6, line 86, “effective life-quality campaign”, please be specific

Page 6, line 86, low-income country: country should be plural

Line 96, remove extra space after “[11, 14]”

Line 101, “effect” does not seem to be the best choice of word, consider revising.

Line 111, from the context, low household income is not a risk factor but an associated factor.

Line 125, remove extra space before “with about 323..”

Line 127, add “patient” after “NS”

Line 133-135, the ethics statement is duplicative – move and merge with the later section on ethics approval.

Line 142-151, I suggest listing these descriptions in a table.

Line 222, the link to the website is not necessary as it is in the citation.

Line 223, replace “American” with “US”

Line 237, does “single” here mean “single in marital status” or just one person? Maybe rephrase to avoid confusion.

Line 240, peasant and farmers are repetitive

Author Response: We are thankful the Reviewer for these detections. We have ajusted your specific comments as below:

Text Insertion (if applicable)/ page/ line number of change:

Page 6, line 86, “effective life-quality campaign”, please be specific.

Response: we added an additional part to be specific to the “effective life-quality campaign”

(Page 5, line 148-149) “..to reduce the burden imposed on households and communities by NS in low-income countries. This would result in an improvement in the quality of life of such household and communities.”

Page 6, line 86, low-income country: country should be plural.

Response: this error has been fixed.

(Page 5, line 150) “..to reduce the burden imposed on households and communities by NS in low-income countries.”

Line 96, remove extra space after “[11, 14]”

Response: the extra spacing has been removed 

Line 101, “effect” does not seem to be the best choice of word, consider revising.

Response: the word “effect” has been replaced with “impact”.

(Page 6, line 169) “..NS is reported to impact a high socioeconomic burden..”

Line 111, from the context, low household income is not a risk factor but an associated factor.

Response: the word “risk factor” has been replaced with “associated factor”

(Page 7, line 194) “It is, therefore, not surprising that low household income has been reported as one of the factors associated with NS”

Line 125, remove extra space before “with about 323..”

Response: The extra space before “with about 323..” has been removed. 

Line 127, add “patient” after “NS”

Response: This part had been adjusted.

Line 133-135, the ethics statement is duplicative – move and merge with the later section on ethics approval.

Response: The ethics approval statement has been removed from line 133 -135 and merged to the later section on ethics approval (Page 9, line 282-286)

Line 142-151, I suggest listing these descriptions in a table.

Response: Descriptions of item on Lines 142-151 have been put in a table labeled Table 1 and the paragraph from line 142-151 has been adjusted as below:

(Page 8, line 232-238) Activities of daily living (ADL), instrumental activities of daily living (IADL) and household activities of daily living (HDL) of caregivers were assessed using the ‘monetary valuation of informal care using the opportunity cost and proxy good method sections’ of the iMTA Valuation of Informal Care Questionnaire (iVICQ) [29]. However, the HDL and ADL questions have been adapted to the Ugandan situation based on a survey by Oxfam in the Philippines, Uganda, and Zimbabwe [30]. A detailed description of the ADL, IADL and HDL activities is presented in Table 1.

(Page 27)

Table 1. ADL, IADL and HDL with their descriptions.

Activity group Specific activities 

Activities of daily living (ADL) Aiding patient with personal care 

Aiding patient in visiting the toilet

 Aiding patient in moving around within the house

Aiding patient with eating and drinking

Instrumental activities of daily living (IADL) Aiding patient in travelling outside the house

Aiding patient in making trips and visiting family or friends

Aiding patient in visiting a doctor or the hospital

Aiding patient in organizing help, physical aids or house adaptations

Aiding patient in taking care of financial matters like buying and selling things, etc.

Household activities of daily living (HDL) Aiding patient in food and drink preparation

Aiding patient in washing the dishes

Aiding patient in cleaning the house or compound

Aiding patient in washing, mending, ironing clothes

Aiding patient in taking care of and playing with your children

Aiding patient in shopping household supplies (incl. food) 

Aiding patient in other tasks like grinding and pounding, collection of water and fuel (e.g. firewood, charcoal)

Line 222, the link to the website is not necessary as it is in the citation.

Response: the link to the website has been removed and the entire section has been moved to the discussion section

Line 223, replace “American” with “US”

Response: we replaced “American” with “US” and this is now in the section that has been moved to the discussion part

Line 237, does “single” here mean “single in marital status” or just one person? Maybe rephrase to avoid confusion.

Response: the word “single” mean “one person”. We rephrased it as below:

(Page 12, line 404) “There were 14 caregivers recruited for this study and except for one male caregiver”

Line 240, peasant and farmers are repetitive

Response: we removed the word “peasant” 

(Page 12, line 407) “All respondents were farmers.”

******

Reviewer #2: Summary: This paper examines the economic burden of nodding syndrome on affected households in Northern Uganda using a cross-sectional survey data. This paper studies an interesting topic that may have important policy implications. It has strength such as having a unique data and conducting thorough literature review. However, it presented major problems that prevented me from recommending publication. My major concerns and comments are the following:

Comment 1) Title

The title is a little misleading as the paper actually examines the economic burden of NS on the caregiver. Household also includes the patient, but the cost doesn’t not consider any cost directly related to NS patients.

Author Response: We are thankful for your kind correction. Based on your suggestion, we have rephrased the study title as “Economic burden of the persistent morbidity of nodding syndrome on caregivers in affected households in Northern Uganda”.

Text Insertion (if applicable)/ page/ line number of change:

(Page 1, line 1-2) “Economic burden of the persistent morbidity of nodding syndrome on caregivers in affected households in Northern Uganda.”

Comment 2) Introduction

The authors didn’t demonstrate a strong motivation of this paper. Line 104 to 111 discussed a little bit but it is not strong enough. Any serious child disease, which requires constant caregiving, could fit in the context the authors discussed. Why looking at economic burden of NS is important? How the cost estimate information could be used for policy makers?

Author Response: We appreciate your kind and constructive comment. The introduction part has been revised and adjusted to enhance the motivation of our paper by focusing on the economic burden and how the cost information could be used by policy maker.

Text Insertion (if applicable)/ page/ line number of change:

(Page 6, line 169-172)

“...for example, such patients are often reported to drop out of school and also, when left untreated, they may not be able to contribute to family income in terms of working, as well as other household duties [19]..”

(Page 7, line 189-193)

“..This situation is further exacerbated by the fact that NS occurs in poverty prone areas. For instance, for the case of Uganda, NS occurs in the northern region. Having undergone a civil war in the past, the north is the most affected region in the country suffering from a double burden of high poverty levels [24,25] and the NS epidemic [26]..”

(Page 7, line 199-201)

“..so as to guide policy makers on how best to support such affected communities in the most resource efficient way in order to alleviate the burden imposed on them by NS.”

Comment 3) Methods

Study population

• The study was conducted from 1/2019 to 2/2019. When did the direct cost occur? Were the direct costs adjusted for inflation? If so, which year where they adjusted for?

Author Response: We are thankful to the Reviewer’s comment and totally agree to this remark. The direct costs were retrospectively assessed over a 1-month period, preceding the data collection. The direct costs were also adjusted for inflation for the year 2018. The adjustment was done as follow: direct costs were converted to USD at an exchange rate of 1 USD = 3704 Ugandan shillings [43], and then inflated to 2018 figures using the Consumer Price index (CPI). All indicators used in these conversions were from the World Bank Official website [43]. Additional paragraphs has been added to clarify this issue.

Text Insertion (if applicable)/ page/ line number of change:

(Page 9, line 288-289) “Costs were assessed over a 1-month period, preceding the data collection. They were analyzed from the caregiver perspective. The direct costs were also adjusted for inflation for the year 2018...”

(Page 11-12, line 387-400) “..These were then weighed by the proportion of people who indicated having foregone unpaid work to provide informal care. Direct and indirect costs were converted to USD at an exchange rate of 1 USD = 3704 Ugandan shillings. For meaningful comparison to studies from other countries, we inflated all local currencies of the study year to 2018 figures using the Consumer Price index (CPI) and then later converted to 2018 international dollars ($) using the power purchasing parity (PPP) for private consumption. All indicators used in these conversions were from the World Bank Official website [43].”

Comment 4) 14 caregivers and 24 participants were in the sample. How were the participants selected? Was the selection random? What is the selection criteria?

Author Response: Dear Reviewer, we appreciate your valuable comments for the sample size. It’s right and exact. Regarding the criteria that we have used in identifying the households, this study was performed in a village locating at Northern Uganda (Lekwala). The inclusion criteria consisted of the households with nodding syndrome patients, who are members of a community-based organization called Alliance for Community with Nodding Syndrome (ACNS), which is being supported by Uganda-Japan Nodding Syndrome Network. After screening, we identified 19 households who meet the inclusion criteria and among them, five were excluded due to translocation to other districts. There were finally 14 households who enrolled to our study. An additional part has been added into the manuscript to clarify the inclusion criteria. 

Text Insertion (if applicable)/ page/ line number of change:

(Page 8, line 217-223)

“..The inclusion criteria consisted of the households with nodding syndrome patients, who are members of a community-based organization called Alliance for Community with Nodding Syndrome (ACNS), which is being supported by Uganda-Japan Nodding Syndrome Network. After screening, we identified only 19 households who meet the inclusion criteria in this village and among them, five were excluded due to translocation to other districts. There were finally 14 households who enrolled to our study.

Comment 5) Data analysis.

• Costs were analyzed from the caregiver perspective. Did you assume that caregiver pay for the costs? Is it possible that caregiver only provides care but not pay for the cost? If so, the wording needs to be revised.

Author Response: Dear Reviewer, thank you for this observation. You raised an important point here. In our study, we considered the caregiver who mainly bore the costs. Among 14 caregivers, 6 of them were either divorced, separated or widowed and therefore they were responsible for paying the cost of their respective households. However, we also acknowledge that other households’ members might have contributed to this activity to a certain degree. Unfortunately, data regarding sharing the financial burden of secondary caregivers such as other members of the extended family were hardly retrieved and then they are unavailable. Given this, the exclusion of secondary caregivers in our study would result to underestimation of the cost. Thus, we have acknowledged it as a limitation and an additional part has been added to clarify this issue. 

Text Insertion (if applicable)/ page/ line number of change:

(Page 20, line 745-750)

“..Since the primary caregivers had not obviously spent their whole time in caring these patients, secondary caregivers such as other members of the extended family as well as neighbors might have contributed to this activity to a certain degree. Therefore, we therefore acknowledge that the failure to assess the indirect and even direct cost burden sharing by other family members and neighbors in our study would result to underestimation of the cost..”

Comment 6) The authors should provide more details regarding how the direct costs were calculated

Author Response: We are thankful to the Reviewer’s comment and totally agree to this remark. In following your suggestion, we have added an additional paragraph in the limitation part of our study. There has been elaboration on direct costs computation as direct costs were computed by adding the expenditure on items under direct medical costs to the ones under direct non-medical costs and then weighted by the proportion of respondents who indicated having spent on such items under each category.

Text Insertion (if applicable)/ page/ line number of change:

(Page 10, line 347-350)

“..Direct costs were computed by adding the expenditure on item(s) under the direct medical costs to the ones under direct non-medical costs weighted by the proportion of people who indicated having spent on such item(s) under each category.”

(Page 13, line 447-451)

“Direct cost of NS 

The monthly weighted mean direct cost per NS patient was 2.2 USD (with a weighted median cost of USD 1.3) corresponding to an annual cost of 26.4 USD (Table 6). Direct cost accounted for about 7.7% of household expenditure, with more than half of the cost incurred on travel and cumulatively, over 90% primarily incurred on direct non-medical costs.”

Comment 7) Line 200, should be “times” not “plus”

Author Response: We agree with this detection and we have corrected this mistake.

Text Insertion (if applicable)/ page/ line number of change:

(Page 10, line 362)

“..the real national median monthly wage (gender-and age-specific) times the labor force participation rate (39.6%) and employment rate (90.6%) for the Acholi sub-region.”

Comment 8) The wage rate used to calculate the indirect costs is the real median monthly wage. Does this wage take into account age and sex factor? If not, in this sample, the majority are women, using a generic wage rate would not be appropriate if there is a substantial gender wage gap. The authors may consider using a weighted average wage based on the gender proportion in the sample.

Author Response: We are thankful to the Reviewer’s comment and totally agree to this remark. Indeed, men actually earn twice as compared to women [37] and therefore, to account for this gender wage gap, instead of using the real monthly wage rate, we used the gender-and age-specific median national wages to estimate the indirect costs and then weighted by the number of caregivers who indicated having foregone time to provide care to NS patients. The data has been re-analyzed based on the new gender-and age-specific median national wages. Some additional parts had been added and the new analysis had been represented in Table 6 to clarify this issue.

Text Insertion (if applicable)/ page/ line number of change:

(Page 10, lines 359-361)

“…and it has been reported that men on average earn twice (134,042 Ugandan shillings; ~37.2 USD) as compared to women (67,021 Ugandan shillings; ~18.6 USD), a proxy was used in computing the informal care costs: the real national median monthly wage (gender-and age-specific) times the labor force participation rate (39.6%) and employment rate (90.6%) for the Acholi sub-region”.

(Page 10, line 361-363)

“…a proxy was used in computing the informal care costs: the real national median monthly wage (gender-and age-specific) times the labor force participation rate (39.6%) and employment rate (90.6%) for the Acholi sub-region…”

(Page 13, line 452-457)

“Indirect cost of NS

The indirect cost was caregivers’ valued (unpaid care) time devoted to taking care of NS patients. This time accounted for less than half (~ 40%) of all forgone care time as leisure time was not valued. Therefore, the average ± SD (standard deviation) caregiving time per week was estimated at 2.4 (± 0.7) hours with a total of 14.5 hours (Table 5). The monthly weighted mean indirect cost per NS patient was 0.1 USD (and zero weighted median cost) (Table 6).”

(Page 13, line 458-461)

“Total cost of NS

The weighted mean monthly sum of direct and indirect costs per NS patient was estimated at 2.3 USD (96.2% direct cost, 3.8% indirect cost) (Table 6), corresponding to an annual cost 27.6 USD.The total weighted median monthly cost has been estimated at 1.4 USD.” 

(Page 35-36)

Table 6. Direct and indirect (informal caregiving) costs of NS 

 Cost Category No. (%) households who incurred direct and indirect costs Mean (SD)

Monthly amount

(USD) Weighted mean monthly amount (US$) Median (IQR) monthly amount (US$) Cost profile (%)

DIRECT COST 

 Medicala 

 Booklet purchase 13 (92.9) 0.1 0.0 0.1 0.0 (0.0 – 0.1) 2.4

 Non-medicalb 

 Meals 11 (78.6) 1.1 2.3 0.8 0.3 (0.3 – 1.0) 35.9

 Travelc 14 (100.0) 1.3 0.8 1.3 1.1 (0.5 – 1.6) 57.9

Total Direct Costs – – 2.5 (3.1) 2.2 1.4 (0.8 – 2.6) 96.2d

INDIRECT COSTSe 

 Valued foregone time devoted to informal caregiving 6 (42.9) 0.2 (0.1) 0.1 0.1(0.1 – 0.2) 3.8

Total Indirect Costs – – 0.2 (0.1) 0.1 0.1(0.1 – 0.2) 3.8

Total costs* – – 2.7 (3.2) 2.3 1.5(0.9 – 2.8) 100.0

Weighted median direct cost was USD 1.3 (0.929*0.0 + 0.786*0.3 + 1*1.1) and indirect costs was USD 0.0 (0.429*0.1).

Note: 1 USD = 3,704 Ugandan shillings; SD: standard deviation; IQR: interquartile range and there were 24 patients.

a Treatment is being provided free by the government of Uganda.

b Only 2 households incurred accommodation expenditure; (one incurred accommodation costs worth 75 USD, more than all direct costs combined (53.55 USD) and the other 4.2 USD, giving an average of 39.6 USD for these two households).

c Travel costs are being supported by a local community-based organization in the area.

d Direct costs account for 7.7 % of household expenditure (excluding the amount for accommodation).

e Only foregone unpaid work time (approximately (~) 40% of all foregone time) was valued, leisure time was not valued.

f Direct and indirect costs were computed per NS patient

Comment 9) The discussion of the relation of informal caregiving cost to GDP could be moved to the discussion as it is not part of the cost calculation. (line 222 to 228)

Author Response: We are thankful for this valuable suggestion. This section has now been moved to the discussion section (Page 18, line 668-676).

Comment 10) The authors didn’t provide a monetized value of intangible cost but instead provided caregiving burden. I wonder if the authors may find a way to obtain measures similar to the monetized QALY to capture the intangible costs? If not, it doesn’t make a lot of sense to show it along with direct cost and indirect cost.

Author Response: Thank you for pointing out this important remark. We totally agree with your suggestion and we have removed this section. An additional paragraph had been added to clarify this issue.

Text Insertion (if applicable)/ page/ line number of change:

(Page 9-10, line 291-345)

“However, intangible costs which constitute an essential part of cost burden for a health condition have not been considered in this study. Intangible costs are mainly the non-physical costs that a patient(s) and their families experience due to ill health or while undertaking treatment and are difficult to measure and assess, for instance the pain and suffering associated with the disease. They focus on caregivers’ subjective burden, health and well-being assessed from their perspective and as have been stressed by Hoefman et al. al. [29], the measurement of subjective burden is considered a non-monetary valuation.” 

Comment 11) Line 233, was the alpha calculated based on the current data? If so, the sample size may be too small to be trusted.

Author Response: Dear Reviewer, this section fell under the one assessing intangible costs and has been removed. 

Comment 12) Results

• I am wondering how severe of the NS patients are in this sample? Is it possible for authors to present the average seizure severity score of this country patients? It is possible that patients with different severe level of NS may require different level of care which may affect the direct costs.

Author Response: Thank you so much for poiting out this important remark. It’s a great idea since patients with different severe level of NS may require different level of care which may affect the direct costs. Indeed, all patients get their medications on a monthly basis from the health care facility and follow a daily regimen to control the seizure and other symptoms; supervised by community member, but patients who did not take the medications (who were said to be allergic to the medication according to the understanding of caregivers) were reported to have severe conditions. We have presented the average seizure severity score in our manuscript at Table 4 (page 31) and unfortunately, we can not reach in calculating the direct cost according to different level of severity. In fact, the distribution of patients’ severity scores in the current study with the range of Hague severity scale range (13-53) suggests that, overall, the majority of the patients experienced average seizure severity. Additionally, only a handful of the patients had daily seizures. 

Comment 13) The sample size is very small which may cause inaccuracy of the estimates of direct costs and time of spending on informal care giving.

Author Response: We agree with this remark. The sample size of our research, which is a cross-sectional study, is unfortunately very small. Therefore this is essentially a pilot study and we have acknowledged this restrictions in the limitation part of our manuscript as below

(Page 20-21, line 754-760)

“..Another limitation of our research is that it is a cross-sectional study with a small sample size and is thus not generalizable to all NS households”

Comment 14) Line 254, this subtitle is confusing. Consider changing it to indirect cost

Author Response: Dear Reviewer, this subtitle has changed to indirect cost based on your suggestion. (Page 13, line 452) 

Comment 15) Line 270-279, there is no monetized value for the intangible cost, and thus not comparable to other costs. Considering removing this part.

Author Response: Dear Reviewer, thank you for this suggestion. This section has been removed. 

Comment 16) Tables

• Table 5 doesn’t seem to be directly relevant to the cost calculation.

Author Response: Dear Reviewer, table 5 is not relevant to the cost computation and has been removed based on your recommendation.

---

## [Decision Letter · Decision Letter 1]

21 Aug 2020

Economic burden of the persistent morbidity of nodding syndrome on affected households in Northern Uganda.

PONE-D-20-04251R1

Dear Dr. Huy,

We’re pleased to inform you that your manuscript has been judged scientifically suitable for publication and will be formally accepted for publication once it meets all outstanding technical requirements.

Kind regards,

Xiangming Fang

Academic Editor

PLOS ONE

Reviewers' comments:

Reviewer's Responses to Questions

**Comments to the Author**

1. If the authors have adequately addressed your comments raised in a previous round of review and you feel that this manuscript is now acceptable for publication, you may indicate that here to bypass the “Comments to the Author” section, enter your conflict of interest statement in the “Confidential to Editor” section, and submit your "Accept" recommendation.

Reviewer #1: All comments have been addressed

Reviewer #2: All comments have been addressed

2. Is the manuscript technically sound, and do the data support the conclusions?

Reviewer #1: Yes

Reviewer #2: Yes

3. Has the statistical analysis been performed appropriately and rigorously? 

Reviewer #1: Yes

Reviewer #2: N/A

4. Have the authors made all data underlying the findings in their manuscript fully available?

Reviewer #1: Yes

Reviewer #2: Yes

5. Is the manuscript presented in an intelligible fashion and written in standard English?

Reviewer #1: Yes

Reviewer #2: Yes

6. Review Comments to the Author

Reviewer #1: The authors have addressed my concerns and I have no further comments. The authors have indicated that the data are available from the authors.

Reviewer #2: Thanks the authors for addressing my comments and revising the manuscript accordingly. All comments have been carefully addressed. My only concern is the small sample size but as the authors mentioned this is a pilot study and they have acknowledged this in the limitation section and the results are not generalizable.

7. PLOS authors have the option to publish the peer review history of their article (what does this mean?). If published, this will include your full peer review and any attached files.

Reviewer #1: No

Reviewer #2: No

---

## [Editor Report · Acceptance letter]

10 Sep 2020

PONE-D-20-04251R1 

Economic burden of the persistent morbidity of nodding syndrome on caregivers in affected households in Northern Uganda. 

Dear Dr. Huy:

I'm pleased to inform you that your manuscript has been deemed suitable for publication in PLOS ONE. Congratulations! Your manuscript is now with our production department. 

Kind regards, 

on behalf of

Dr. Xiangming Fang 

Academic Editor

PLOS ONE